

# Statistical study of ULF waves in the magnetotail by THEMIS observations

**Shuai Zhang[1,2], Anmin Tian[1*], Quanqi Shi[1], Hanlin Li[1], Alexander W. Degeling[1], I. Jonathan Rae[3], Colin Forsyth[3], Qiugang Zong[4], Mengmeng Wang[1], Xiaochen Shen[1], Weijie Sun[5], Shichen Bai[1], Ruilong Guo[6], Huizi Wang[1], Andrew Fazakerly[3], Suiyan Fu[4], Zuyin Pu[4]**

[1]Shandong Provincial Key Laboratory of Optical Astronomy and Solar-Terrestrial Environment, School of Space Science and Physics, Shandong University, Weihai, 264209, China

[2]State Key Laboratory of Space Weather, Chinese Academy of Sciences, Beijing 100190, China

[3]University College London, Mullard Space Science Laboratory, Space and Climate Physics, Dorking, United Kingdom

[4]School of Earth and Space Sciences, Peking University, Beijing 100871, China

[5]Department of Climate and Space Sciences and Engineering, University of Michigan, Ann Arbor, USA

[6]Institute of Geology and Geophysics Chinese Academy of Sciences, Beijing 100029, China

*Correspondence to*: A. M. Tian (tamin@sdu.edu.cn)

**Abstract.** Ultra-low frequency (ULF) waves are ubiquitous in the magnetosphere. Previous studies mostly focused on ULF waves in the dayside or near-earth region (with radial distance $R < 12\ R_E$). In this study, using the data of Time History of Events and Macroscale Interactions during Substorms (THEMIS) during the period from 2008 to 2015, the Pc5-6 ULF waves in the tail region with $X^{*}_{GSM} < 0$, $8\ R_E < R < 32\ R_E$ (mostly on the stretched magnetic field lines) are studied statistically. A total of 1089 azimuthal oscillating events and 566 radial oscillating events were found. The statistical results show that both the azimuthal and radial oscillating events in the magnetotail region ($12\ R_E < R < 32\ R_E$) are more frequently observed in the post-midnight region. The frequency decreases with increasing radial distance from Earth for both azimuthal oscillating events ($8\ R_E < R < 16\ R_E$) and radial oscillating events ($8\ R_E < R < 14\ R_E$), which is consistent with the field line resonances theory. About 52 % of events (including the azimuthal and radial oscillating events) are standing waves in the region of 8-16 $R_E$, while only 2 % are standing waves in the region of 16-32 $R_E$. There is no obvious dawn-dusk asymmetry of ULF wave frequency in $8\ R_E < R < 32\ R_E$, which contrasts with the obvious dawn-dusk asymmetry found by previous studies in the inner magnetosphere ($4\ R_E < R < 9\ R_E$). An examination for possible statistical relationships between ULF wave parameters and substorm occurrences is carried out. We find that the wave frequency is higher after the substorm onset than before it, and the frequency differences are more obvious in the midnight region than in the flank region.



**Keyword.** Magnetospheric physics (Magnetotail; MHD waves and instabilities; Solar wind-magnetosphere interactions)

**1 Introduction**

Ultra-low frequency (ULF) waves play a significant role in storing and transferring energy in the Earth's magnetosphere. ULF waves can transport energy from the magnetosphere to the ionosphere, accelerate energetic particles, modulate luminosity of aurorae, mediate reconnection and trigger substorm onset (e.g., Baumjohann and Glassmeier, 1984; Lessard et al, 1999; Ukhorskiy et al., 2005; Keiling, 2009; Rae et al., 2014; Zong et al., 2017).

There are several excitation sources for magnetospheric ULF waves. These sources include the Kelvin-Helmholtz instability (KHI) along the magnetopause (e.g., Walker, 1981; Claudepierre et al., 2008), solar wind dynamic pressure impulse (e.g., Allan et al., 1986; Lee et al., 1989; Shi et al., 2013; Degeling et al., 2014; Shen et al., 2015), periodic solar wind dynamic pressure variations (e.g., Kepko,2002; 2003), drift-bounce resonance (e.g., Southwood et al.,1969) and dynamic processes during substorms (e.g., Olson, 1999).

Although many previous studies have focused on waves occurring in the dayside magnetosphere (e.g., Samson et al., 1981; Rostoker et al., 1984; Zong et al., 2007; Shen et al., 2017), ULF waves occurring on stretched magnetic field lines in the magnetotail have also been reported in some observational studies (e.g., Zheng et al., 2006; Tian et al., 2012) and simulations (e.g., Rankin et al., 2000; Lui and Cheng, 2001). Pc5 (150-600 s) and Pc6 (600-1800 s) waves are the primarily waves occurring at high latitudes and in the magnetotail. Investigating the source and characteristics of these waves in the magnetotail will help us further understand the solar wind-magnetosphere-ionosphere coupling processes in the night side region.

Statistical studies of ULF wave properties in the magnetosphere have been performed using various satellites (e.g., Hudson et al., 2004; Liu et al., 2009; Takahashi et al., 2015). Hudson et al. (2004) performed a statistical study of the occurrence rate of Pc5 magnetic pulsations for both toroidal and poloidal modes at L values from 4 to 9 by using 14 months magnetometer data from Combined Release and Radiation Effects Satellite (CRRES). They found that there is no dawn-dusk asymmetry on the occurrence rate of toroidal mode oscillations inside L=8, however the occurrence rate of poloidal mode oscillations is higher on dusk side. Liu et al. (2009) statistically studied the both occurrence and frequency distributions of Pc5 magnetic pulsations in toroidal and poloidal modes between L= 4 and 9 by using 13 months electric and magnetic field measurements from THEMIS-D. They found that the occurrence distribution is similar to the results of Hudson et al. (2004) and the frequency is higher in the dawn side than in the dusk side by a factor 2 and decreases with radial distance. Takahashi et al. (2015) statistically investigated the fundamental toroidal mode oscillations from L = 7 to 12 by using 2008-2013 ion bulk



velocity data from THEMIS-D. They found that the occurrence rate and amplitude of toroidal mode oscillations are higher in the dawn side (4-8 MLT) than in the dusk side (16-20 MLT). Moreover, the relationship between ULF wave characteristics and the solar wind conditions/geomagnetic activity level were also studied statistically (e.g. Takahashi and Ukhorskiy. 2007; Susumu Kokubun, 2013; Wang et al., 2015). Takahashi and Ukhorskiy (2007) found that the solar wind dynamic pressure variance has the best correlation with the power of magnetic pulsations at geosynchronous orbit. Susumu Kokubun (2013) statistically studied Pc5 ULF waves (mostly on the 4-8 MLT and 16-20 MLT) using GEOTAIL data during the period of 1995 to 2000. They found that the wave occurrence tends to be larger for higher solar wind velocity (> 400 km/s), smaller IMF Bz, and lower cone angle. Wang et al. (2015) studied the spatial distribution of the Pi2 and Pc4-5 magnetic fluctuation power in the plasma sheet by using THEMIS-A/C/D/E data from 2007 to 2014. They found that the amplitude of Pc-5 fluctuations is larger globally during periods of higher AE index, faster solar wind, and larger solar wind dynamic pressure variations.

Although statistical studies of ULF waves have been performed, most have focused on the dayside or near-earth region. The distributions and excitation mechanisms of ULF waves on stretched magnetic field lines are still unclear. Our work focuses on ULF waves on stretched magnetic field lines ($X^*_{GSM} < 0$ and $8\ R_E < R < 32\ R_E$).

This paper will be organized as follows. In section 2, the data set and the selection criteria of the ULF wave event are presented. In section 3, we show the statistical results. In section 4, we discuss the occurrence and frequency distributions of ULF waves on the stretched field lines and the influence factors of solar wind parameters and geomagnetic activity level. The main conclusions of this study are given in section 5.

## 2 Data and statistical methods

In this study, we use 3 s resolution magnetic field data from Flux Gate Magnetometer (FGM) (Auster et al., 2008) and 3 s resolution plasma data from Electrostatic Analyzer (ESA) (Mcfadden et al., 2008) of THEMIS mission from 2008 to 2015. The THEMIS mission consists of five satellites (THEMIS A/B/C/D/E), each with an orbital inclination of about 10° (Angelopoulos, 2008). In the first two years, the apogees were about 12 $R_E$ for THEMIS A/D/E, 20 $R_E$ for THEMIS C and 30 $R_E$ for THEMIS B. After 2010, THEMIS B/C were transferred to a lunar orbit which is about 60 $R_E$ from earth. Because THEMIS A/D/E have similar orbits, in this study we only use data from THEMIS A, B and C. In addition, we use 1 minute resolution interplanetary magnetic field (IMF) and solar wind plasma data from the OMNI database (https://spdf.sci.gsfc.nasa.gov/), which is calculated by time shifting satellite data taken in the solar wind to the Earth's bow shock sub-solar point. Figure 1 shows the binned spatial distribution of the total observation time over the 2008-2015 interval for THEMIS A/B/C in the magnetosphere.

We use the aberration coordinate GSM$^*$ whose X axis rotated 4° from the X axis of GSM coordinates for spacecraft position to remove the effect of Earth's revolution. Field-aligned coordinates



(FAC) are used to analyze waves and separate the azimuthal and radial oscillating wave components. The
FAC system is defined in Eq. (1).

$$\mathbf{z} = \frac{\mathbf{B}_0}{|\mathbf{B}_0|}; \mathbf{a} = \frac{\mathbf{z} \times \mathbf{R}}{|\mathbf{z} \times \mathbf{R}|}; \mathbf{r} = \mathbf{a} \times \mathbf{z} \quad (1)$$

In this equation, $\mathbf{B}_0$ is the background magnetic field vector, derived by taking a 30 minutes sliding
average of the magnetic field data, $\mathbf{R}$ is the vector from Earth's center to the satellite, $\mathbf{z}$ is the parallel unit
vector, $\mathbf{a}$ is the unit vector pointing east and $\mathbf{r}$ completes the right-hand rule. It should be noted that the
direction of $\mathbf{r}$ is approximately radial due to the equatorial orbits of THEMIS.

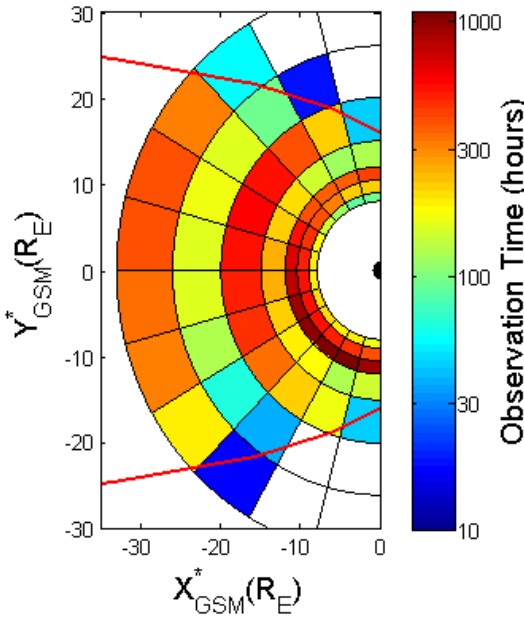

**Figure 1.** The distribution of total observation time of THEMIS A/B/C in the GSM* X-Y plane between
2008 and 2015. The red line is the average magnetopause, calculated by Shue et al.'s (1998) model with
Dp=1.66 npa and Bz=0.16 nT. The blank bins indicate regions where the residence time of THEMIS is
less than 10 hours.


In this study, we mainly use ion velocity data to identify ULF waves, following the technique of
Takahashi (2015). They suggested that using velocity is better than using magnetic field data, because
fundamental mode magnetic field fluctuations (considered most likely in the Pc5 range) give rise to a
node near the equatorial plane, making their measurement problematic along the low-inclination THEMIS
orbital path. On the other hand, the fundamental mode has an antinode for the electric field and plasma
velocity fluctuations under ideal MHD conditions. The electric field data is therefore estimated by $\mathbf{E} =$
$-\delta\mathbf{V} \times \mathbf{B}$, where $\delta\mathbf{V}$ indicates the variation of velocity, which is obtained by subtracting the 30 minutes
sliding average values.



As shown in Fig. 1, the region concerned in this work is $X^*_{GSM} < 0$ $R_E$ and 8 $R_E < R < 32$ $R_E$. In order
to remove the likelihood of identification of ULF wave events when THEMIS enters the magnetosheath
or solar wind regions, only events for which density values less than 1 cm$^{-3}$ if $|Y^*_{GSM}| > 10$ $R_E$ are included
in the database.
The following criteria are used to select ULF waves in the magnetotail: (i) the wave frequency is
below 7 mHz; (ii) the wave is quasi monochromatic, and includes at least three cycles; (iii) the maximum
of peak to trough value of fluctuations is more than 50 km/s; (iv) mirror-like structures, indicated by anti-
phase variations of magnetic field and density are excluded; (v) magnetotail flapping events, characterized
by sign changes in Bx are excluded. A quantitative standard is used to determine the beginning and ending
time of each event, namely that the beginning and ending time is at the points where the amplitude is 20
km/s. Additionally, if the interval time between two events is less than 20 minutes and they have similar
frequency (within 0.5 mHz), we consider them as a single event.
The process of selecting wave events and distinguishing the wave mode in this study is as follows.
Firstly, we conduct wavelet analysis to THEMIS ion velocity and magnetic field data in GSM coordinates
and choose the wave events which roughly satisfy criteria mentioned above. Then, we transform from
GSM to FAC coordinates for magnetic field and ion velocity data, and calculate the electric field. To
quantitatively distinguish the azimuthal or radial oscillating waves, Fast Fourier Translation (FFT)
analysis is applied to all three components of ion velocity (Fig. 2).
Figure 2 shows two typical events (labelled "A" and "B") with Event A occurring near R≈19 $R_E$,
from 0550 to 0650 UT on 01 February 2009 and showing azimuthal oscillations, and Event B occurring
near R≈8 $R_E$ from 0728 to 0828 on 11 April 2013 and showing radial oscillations. Figure 2 shows three
components of the ion velocity (a-c), magnetic field (d-f), and the calculated electric field (g-i), in addition
to the total ion density (j) and total magnetic field (k) which are used for excluding mirror-like structures.
Figure 2(l-n) show the Power Spectral Density (PSD) of three components of the ion velocity derived by
FFT. In events A and B, the peak in PSD of the dominant wave component exceeds its counterpart by a
factor of 4, enabling their unambiguous designation as an azimuthal and radial oscillation event
respectively. Events for which the peak in PSD in Va and Vr have similar magnitudes are simply regarded
as both an azimuthal oscillating event and a radial oscillating event.
In total, we find 1089 azimuthal oscillating wave events and 566 radial oscillating wave events, with
an average event-time duration of ~54 minutes.

**3 Statistical Analysis**

**3.1 Occurrence rate**

Figure 3 shows the occurrence rates of azimuthal oscillating wave events (left panel) and radial oscillating
wave events (right panel) in the GSM$^*$ X-Y plane. The color in each bin indicates the occurrence rate





**Figure2.** Examples of an azimuthal oscillating event (Event A) from 0550 to 0650 UT on 01 February 2009, and radial oscillating event (Event B) from 0728 to 0828 on 11 April 2013: (a-c) velocity components, (d-f) magnetic components, (g-i) electric field components, (j) total ion density, (k) total magnetic field, (l-n) FFT analysis of ion velocity.



calculated by dividing the total event times by the total observation times shown in Fig. 1. In the near-
earth region (8 $R_E$ < R < 12 $R_E$), we can see that the occurrence rates of both azimuthal and radial
oscillating events in the dusk and dawn flanks (18-21 MLT and 3-6 MLT) are higher than the midnight
regions (21-03 MLT). For azimuthal oscillating events, there is no clear dawn-dusk asymmetry in the
occurrence rates, while for the radial oscillating events, the occurrence rates of waves are higher on the
dusk side than dawn side. In the magnetotail region (12 $R_E$ < R < 32 $R_E$), the occurrence rates of both
modes of waves are slightly higher in the post-midnight region. Note that, although no wave events are
found in the dawn side flank region (20 $R_E$ < R < 32 $R_E$, 3-6 MLT), the total observation time is also very
short (< 38 hours) in this region. So, we cannot conclusively say that the occurrence rates on the dusk
side flank region is higher than that of the dawn side.

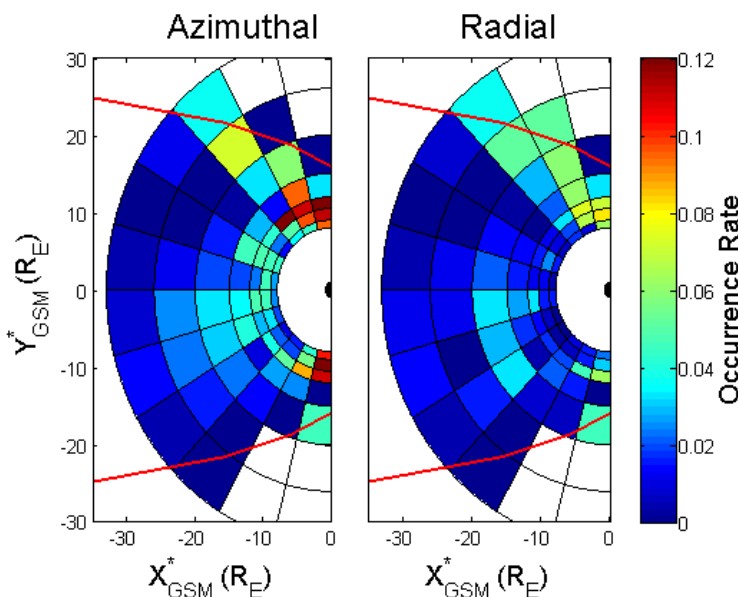


**Figure 3.** The occurrence rates of azimuthal oscillating wave events (left panel) and radial oscillating
wave events (right panel) in the GSM[*] X-Y plane.

**3.2 Frequency distribution**

Figure 4 shows the spatial distribution of average frequency for azimuthal (left panel) and radial (right
panel) oscillating wave events in the equatorial plane. The color in each bin is the average of all event
frequencies (obtained by FFT analysis as described earlier) in that bin. A blank bin inside the
magnetopause indicates that there are no events. It can be seen roughly that the frequency decreases with
increasing radial distance both for azimuthal and radial oscillating wave events, for regions where





R<15R$_E$. Note that the crimson bin in the upper right corner (19-20 MLT and $20 < R < 26$ R$_E$) of the right
panel is caused by short residence time (~19 hours) and only one wave event with frequency of 5.71 mHz.

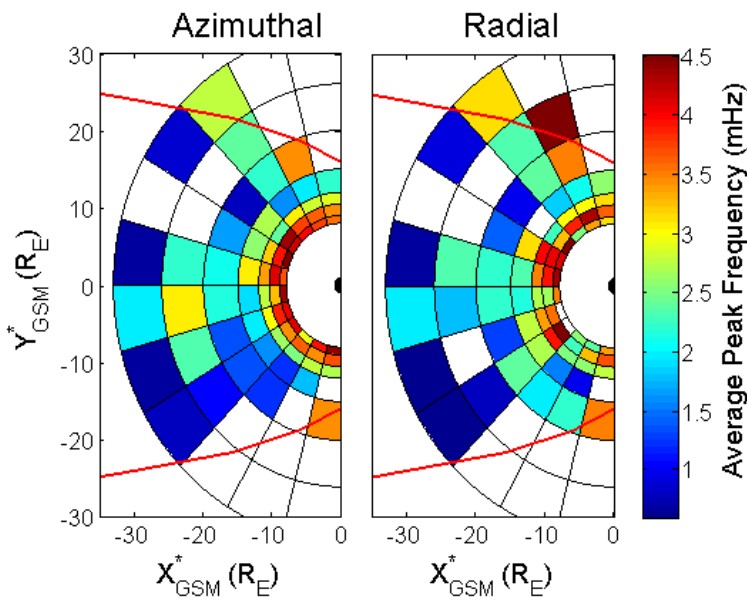


**Figure 4.** The average frequencies of azimuthal oscillating wave events (left panel) and radial oscillating
wave events (right panel) in the GSM$^*$ X-Y plane.


We further plot the relationship between the peak frequency and the distance from earth in Fig. 5. It
shows that the frequency can be as low as 0.55 mHz. As shown in Fig. 5a and 5b, the median frequency
of azimuthal oscillating events decreases with increasing radial distance from the Earth in the region with
8 R$_E$ < R < 16 R$_E$, and the same trend is found for the radial oscillating events with 8 R$_E$ < R < 14 R$_E$.
Figure 5c and 5d show frequency distribution of events in the dawn side (Y$^*$gsm < 0) and dusk side
(Y$^*$gsm > 0) regions, respectively. The frequency for both azimuthal and radial oscillating events show
no obvious dawn dusk asymmetry. This is verified by the Wilcoxon rank sum test applied to the dawn and
dusk datasets. The Wilcoxon rank sum test is a non-parametric statistical hypothesis test that can be used
to assess whether two samples have the same distribution or not (Gibbons and Chakraborti, 2011).
Specifically, in the Wilcoxon rank sum test, a "P-value" result greater than 0.01 means there is no
significant statistical difference between two datasets. The P-value for the dawn and dusk side data sets
is 0.4535 (for all azimuthal and radial oscillating events). This confirms that the dawn and dusk side
frequency data sets belong to the same distribution.





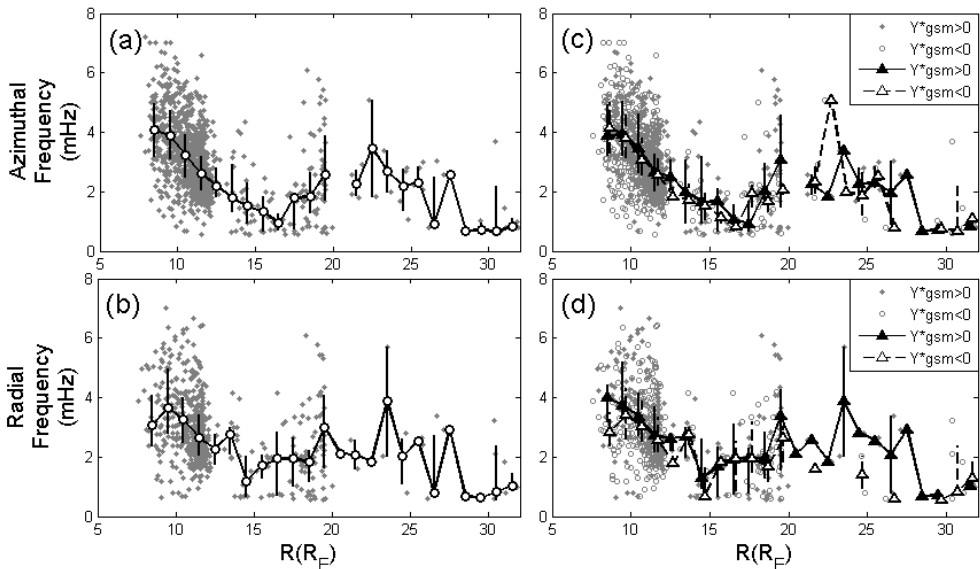

**Figure 5.** The wave frequency versus radial distance for azimuthal (a and c) and radial (b and d) oscillating event. In panels a and b, the grey dots are individual events, the open circles are the median values of frequencies in each 1 $R_E$ bin. The vertical bars connect the lower and upper quartiles. In panel c and d, the grey dots and circles indicate the dusk and dawn events, respectively. The solid and open triangles are the same as the open circles in Fig. 5a, but for dusk and dawn events respectively.

### 3.3 Standing wave

According to Singer et al. (1982), Alfvénic standing wave oscillations are characterized by a phase difference of 90° between the electric field and magnetic field components. Figure 6 shows the standing wave analysis of two azimuthal oscillating events. The first row shows the magnetic field component Ba and electric field component Er. The second row shows the 1.26-3.26 mHz (Fig. 6a) and 2.03-4.03 mHz (Fig. 6b) band-pass filtered Ba and Er components. The phase differences between the band-pass filtered Ba and Er are shown in the bottom panels, in which three dotted lines indicate the 60°, 90°, 120° phase differences respectively. We can see that the first event (Fig. 6a) shows characteristics of standing wave as indicated by the ~90° phase difference between Ba and Er, while the second event (Fig. 6b) does not have this characteristic. We quantify the criteria of standing azimuthal (radial) oscillating waves as that the phase differences between the filtered Ba and Er (Br and Ea) that falls within the range 60°-120° and lasts for at least three cycles.





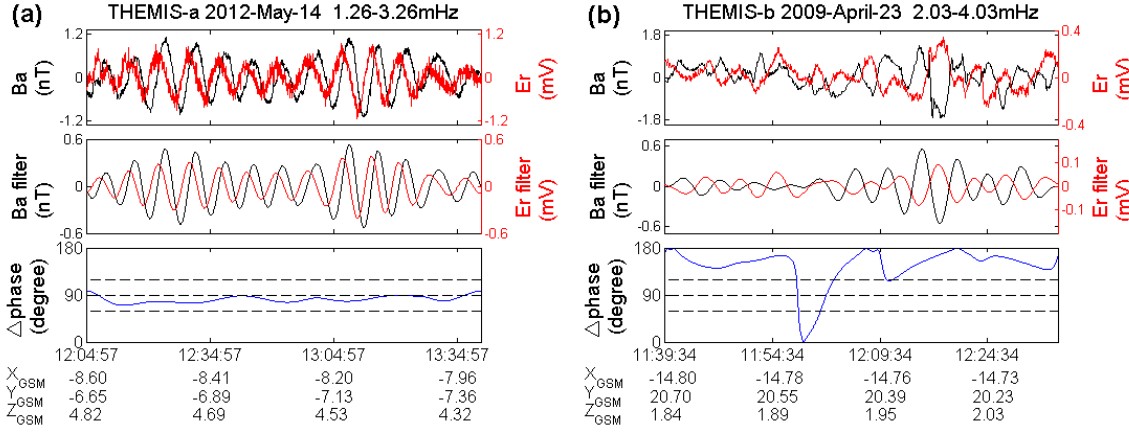

**Figure 6**. Examples of: (a) a standing azimuthal oscillating event and (b) a non-standing azimuthal oscillating event.

Figure 7 shows the distribution of the probability of standing waves with the distance R. The light and deep histogram represents the probability for azimuthal and radial oscillating event respectively. The errorbars shown are calculated by $\varepsilon = \frac{n}{N} * \left( \frac{\sqrt{n}}{n} + \frac{\sqrt{N}}{N} \right)$, where n is the number of standing wave events and N is the total number of waves events in each bin for each polarization. It is obvious that standing waves occupy a larger proportion in the region of 8-16 $R_E$, while almost no standing waves are identifiable in the region of 16-32 $R_E$. We find that about 52% events (including the azimuthal and radial oscillating events) are standing waves in the region of 8-16 $R_E$, while only 2 % are standing waves in the region of 16-32 $R_E$. This figure also shows that the probability of standing waves is higher for the azimuthal oscillating events than for the radial oscillating events.

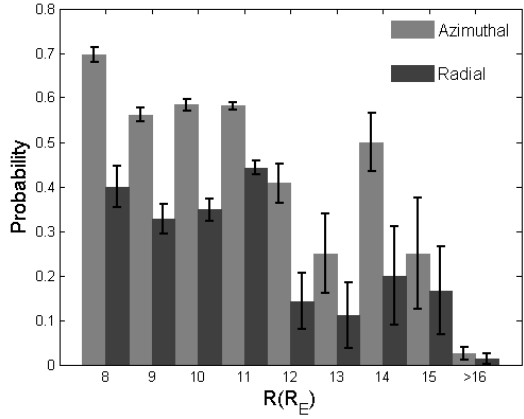

**Figure 7.** The radial distribution of the probability of identifying standing waves, for azimuthal and radial oscillating events (light and dark histograms respectively).

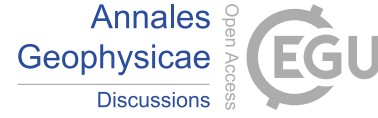




## 4 Discussion



Using THEMIS data during the period from 2008 to 2015, we find 1314 Pc5-6 ULF wave events in the
region of $X^*_{GSM} < 0$ and 8 $R_E$ < R < 32 $R_E$. The elevation angle of the magnetic field of each event was
calculated by the formula $\tan^{-1}\left(\frac{Bz}{\sqrt{Bx^2+By^2}}\right)*\frac{180}{\pi}$, where Bx, By, Bz are the three magnetic field
components in GSM$^*$ coordinates. We find that 58.25% of the events have an elevation angle larger than
45°. This suggests that most of our events are observed near the magnetic equatorial plane. It is reasonable
to consider that most of our standing wave events belong to the fundamental eigenmode. Furthermore,
Lui and Cheng. (2001) indicated that the magnetic field line in the nightside is very stretched in the region
of R > 8 $R_E$, especially during intervals of high Kp index. We therefore consider it likely that most of our
events should be observed on stretched magnetic field lines.

### 4.1 Occurrence rate



As shown in Fig. 3, in the region of 8 $R_E$ < R < 12 $R_E$, there is no obvious dawn-dusk asymmetry in the
occurrence rates for azimuthal oscillating waves, while the occurrence rates are higher on the dusk side
than dawn side for radial oscillating waves. This is consistent with the wave mode distributions in the
inner magnetosphere (4 $R_E$ < R < 9 $R_E$) presented in previous works (Hudson et al., 2004, Liu et al., 2009).
One possible reason is that westward drifting ions injection associated with substorm may excite more
radial oscillating wave events in the dusk side via the ion drift bounce resonance (Southwood et al., 1969;
Chen and Hasegawa, 1988). In contrast to that of the inner magnetosphere (4 $R_E$ < R < 9 $R_E$), the
occurrence rates for both azimuthal and radial oscillating events in the region of 12 $R_E$ < R < 32 $R_E$ are
slightly higher on the post-midnight region than the pre-midnight region. It is possible that the K-H
instability may play an important role on the generation of ULF waves on the stretched magnetotail, given
that the K-H instabilities are inclined to occur in the dawn side than in the dusk side (Nykyri et al., 2013)
and even can happen in the down tail flanks up to the lunar orbit (~60 $R_E$) (Wang et al., 2017). In view of
the limited observation times in the dawn side magnetopause, more events are needed in further study on
the definite reasons of the dawn-dusk asymmetry of occurrence rate in the outer side region (12 $R_E$ < R <
32 $R_E$).

### 4.2 Frequency distribution



As shown in Fig. 5a and 5b, the frequency decreases with increasing radial distance from the Earth for
both azimuthal oscillating events (8 $R_E$ < R < 16 $R_E$) and radial oscillating events (8 $R_E$ < R < 14 $R_E$). This
is consistent with the Alfven continuum of field line resonance (FLR) theory (e.g., Allan and



Poulters.1992; Waters et al., 2000). However, this trend does not continue for R > 16 R$_E$. Previous
observation and simulation studies have shown that standing waves can exist on the stretched magnetic
field lines (Lui and Cheng, 2001; Zheng et al., 2006; Tian et al., 2012). Our statistical result shows that
52 % of all event types are standing waves in the region of 8-16 R$_E$, while only 2 % can be confirmed as
standing waves in the region of 16-32 R$_E$ as shown in Fig. 7. Given the likelihood that most of our wave
events belong to the fundamental mode, the uncertainty in the phase measurement of the weak magnetic
field signal near the equatorial plane will affect the identification of standing waves. Moreover, the
complicated phase relationship between the electric field and the magnetic field caused by magnetic field
disturbances in the farther deeper magnetotail will also affect the identification of standing waves. These
suggest that our data may underestimate the proportion of standing wave events. Even so, the finding that
only 2 % of events in the down-tail region (R > 16 R$_E$) can be identified as standing waves suggests that
the standing waves are far less common on the highly stretched field lines.
As shown in Fig. 5c and 5d, there is no obvious dawn-dusk asymmetry in the ULF wave frequency
for 8 R$_E$ < R < 32 R$_E$. This is different from previous studies in the near-earth region (Liu et al., 2009;
Takahashi et al., 1982; 2015). Takahashi et al. (1982) found that the frequencies of Pc3-4 ULF waves
were higher on the dawn side than dusk side at geosynchronous orbit. They suggested that the quasi-
parallel shock and the associated turbulent magnetosheath flow is more likely to occur on the dawn side,
which leads to higher harmonic waves to be excited in the dawn side. Takahashi et al. (2015) found that
the frequencies of Pc5 toroidal waves in the region with L values between 7 and 12 R$_E$ is lower in the
dusk side (16-20 MLT) than dawn side (04-08 MLT). They suggest that this is due to the higher mass
density in the dusk side near-earth region, supplied by the particles from ionosphere. However, the wave
frequency distributions shown in this paper (X$^*_{GSM}$ < 0, 8 R$_E$ < R < 32 R$_E$) show a different distribution
feature from that of the events in the inner or dayside magnetosphere. This suggests that neither of the
above mechanisms for producing asymmetry are important within the region of interest in our study. This
may be expected for the turbulent magnetosheath flow mechanism more applicable to higher frequencies.
The influence of particle injection from the ionosphere may be weakened by higher ExB drift speeds and
longer field line lengths in the nightside magnetotail region, compared to the near-earth region.

**4.3 The influence of Solar wind parameters and geomagnetic activity level**

Figure 8 shows the relationship between the occurrence rate of wave events and solar wind velocity Vx
(panel a) and the AE index (panel b). The Y-axis indicates the normalized event number, which is
calculated by dividing the number of waves with the proportion of background solar wind velocity Vx
(panel a) and the background AE index (panel b) in each bin. The background values are calculated with
OMNI data from 2008 to 2015. We can see that the ULF waves occurrences increase with increasing solar
wind speed |Vx|. This implies that the K-H instability could be a sources of ULF waves in the magnetotail
region (8-32 R$_E$), since the higher shear velocity is an important factor for exciting K-H instabilities




(Miura, 1992). Figure 8b shows that the waves occurrences are higher when the AE values are less than
500 nT. Note that about 74.8 % of the waves occurred when the AE values are less than 250 nT. This
suggests that most of the wave events in the magnetotail are observed during quiet times or weak substorm
times. The relationship between the occurrences of ULF waves and the relative variation of solar wind
dynamic pressure (Pd) and the IMF Bz values were also examined (not shown). We find that the
occurrence rates are higher for larger solar wind Pd variance and during periods of northward IMF Bz.

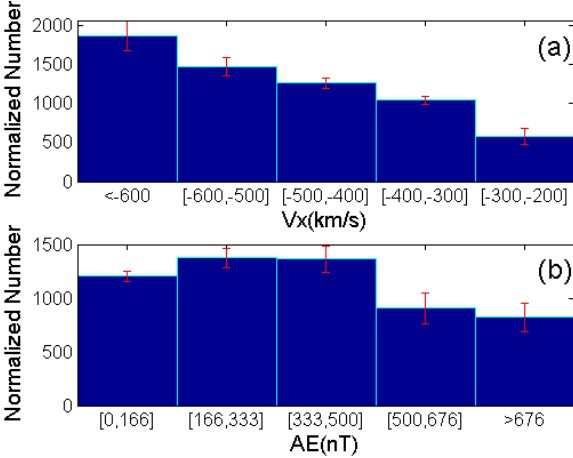


**Figure 8.** The normalized occurrence number of waves versus (a) Solar wind velocity Vx, and (b) AE
index.

334          The possibility that substorm activity may affect the frequency of ULF waves, and thereby influence
the distribution of ULF frequencies in our database, is examined using the following method, based on
the substorm event list of (Forsyth et al., 2015). The ULF wave events were divided into two categories
based on their start time relative to the onset time of individual substorm events. The first category ("type
one") consists of events that occurred more than two hours after the most recent substorm onset, and more
than one hour before the next substorm onset. These events are considered to be independent of substorm
activity. The second category ("type two") consists of events that occurred between zero and two hours
after the most recent substorm onset. In principle a third category consisting of events that occur less than
one hour before the next substorm onset could be defined, however this category contains very few events,
so their frequency characteristics will not be discussed here. The radial dependence of median frequency
for type one and two events is shown in Fig. 9a. This plot clearly shows that the median frequencies for
type two events are higher than type one events. A plausible explanation for this difference could be that
field line depolarization following substorm onset results in an increase in local magnetic field strength
compared to more stretched magnetotail field lines during quiet times. The resulting higher Alfvén speed
profile raises the fundamental mode eigenfrequency for the type two events, compared to the type one
events.



Figure 9b and 9c show the radial dependence of median frequency for type one and two events
occurring in the dawn/dusk flank (3-6 MLT and 18-21 MLT) and midnight sectors (21-03 MLT),
respectively. According to these plots, the frequency differences between type one and type two wave
events are more obvious in the midnight region than in the flank region. This is understandable, given
that the configuration of field lines will be changed much more in the midnight region than in the flank
regions during substorm times. It should be noted that, only the possible influence of field lines
configuration or plasma environment associated with weak substorms on the ULF wave frequencies are
discussed here. The question of whether substorms could trigger or be triggered by ULF waves still cannot
be answered by the present analysis.

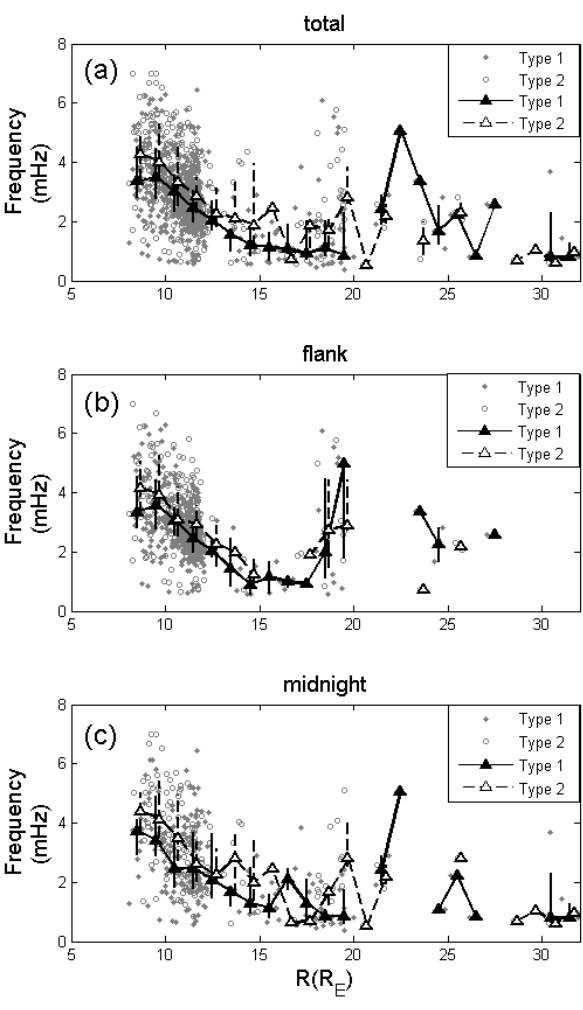


**Figure 9.** The wave frequency versus the distance from the earth for: (a) the type one and type two wave





events, (b) the wave events in the flank region, and (c) the wave events in the midnight region, respectively. The grey dots and circles indicate type one and type two wave individual events respectively. The solid and open triangles are the median values of frequencies in each $1R_E$ bin for the type one and type two wave event respectively. The vertical bars connect the lower and upper quartiles for each category.

## 5 Summary

We have statistically studied the distributions of the occurrence rate and frequency of the Pc5-6 ULF waves in the region of $X^*_{GSM} < 0$ and $8\ R_E < R < 32\ R_E$ (occurring mostly on stretched magnetic field lines) using 8 years of THEMIS data. We also examined the influence of Solar wind parameters and geomagnetic activity level on the features of these ULF waves. Some new results that differ from those of ULF waves observed in the inner magnetosphere are obtained. The main results are summarized as follows:

(1) In the far magnetotail region ($12\ R_E < R < 32\ R_E$), the occurrence rates of both azimuthal and radial oscillating events are higher in the post-midnight region than in the pre-midnight region. In the near-earth magnetotail ($8\ R_E < R < 12\ R_E$), the occurrence rates of azimuthal oscillating events are comparable on the dawn and dusk side, while the occurrence rates of radial oscillating events are higher on the dusk side, which is similar to the distributions in the inner magnetosphere ($4\ R_E < R < 9\ R_E$).

(2) Statistically, the peak frequency decreases as the increase of radial distance from Earth for both azimuthal oscillating events ($8\ R_E < R < 16\ R_E$) and radial oscillating events ($8\ R_E < R < 14\ R_E$). A possible explanation for this distribution is that at least 52 % events (including both azimuthal and poloidal oscillating events) are standing waves in the region of 8-16 $R_E$, while only 2 % are unambiguous standing waves in the region of 16-32 $R_E$. Moreover, the frequencies for all the events in this paper do not show obvious dawn-dusk asymmetry as found in previous studies for waves in the inner magnetosphere ($4\ R_E < R < 9\ R_E$), where the wave frequencies are higher in the dawn side than in the dusk side.

(3) The ULF wave occurrence rates are higher for larger solar wind velocity and solar wind Pd variations. Therefore, we suggest that the solar wind maybe the main energy source of the ULF waves in the region of $8\ R_E < R < 32\ R_E$. About 74.8 % of the ULF waves occurred when the AE values are less than 250nT, which indicates that the ULF waves most likely to occur during in the quiet times or weak substorm times. We have further studied the frequency change between the quiet time and the weak substorm time events. We found that the wave frequency is higher during the substorm time (0-2 hours after substorm onset). The frequency differences are clearer in the midnight region than in the flank region. We suggest that the field lines configuration or plasma environment variation during weak substorm times could increase the eigen frequencies of ULF waves in the magnetotail, leading to the observed change in the frequency distribution.

*Acknowledgments*. We acknowledge THEMIS project team for THEMIS data at



http://themis.ssl.berkeley.edu/, and SPDF web service for OMNI data at https://spdf.sci.gsfc.nasa.gov/.
This work was supported by the Shandong University (Weihai) future plan for Young Scholar
(2017WHWLJH08), the National Natural Science Foundation of China (Grants Nos. 41304129,
41774153, 41574157, and 41628402), the Science and Technology Facilities Council (Grants Nos.
ST/N000722/1), Natural Environment Research Council (Grants Nos. NE/L007495/1, NE/P017150/1 and
NE/P017185/1). Project Supported by the Specialized Research Fund for State Key Laboratories.

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
