# Peer review of "Statistical study of ULF waves in the magnetotail by THEMIS"

_Annales Geophysicae, 2018_

## Referee Comment (RC1) · Anonymous Referee #1 · 8 Jun 2018

**General comments**

This paper presents results from a statistical study of ULF waves in the Pc5-Pc6 frequency range in the magnetotail using magnetic field and plasma data from instruments onboard the THEMIS satellite during 2008–2015. Azimuthally oscillating waves and radially oscillating waves are studied separately. The authors found that in the near-Earth magnetotail the dawn-dusk asymmetry of wave occurrence is observed only in the case of radially oscillating waves. In the far magnetotail, they reveal a higher occurrence rate during post-midnight magnetic local times than during pre-midnight, and that the peak frequency of waves decreases with increasing radial distance from Earth. A majority of events in the near-Earth magnetotail are standing waves, while these represent very small percentage of the events in the far magnetotail. No dawn-dusk

asymmetry could be found in the wave frequencies. Finally, the authors studied the effects of solar wind parameters and geomagnetic activity, and found that ULF wave occurrence is favoured by high solar wind speed, high solar wind pressure variability, but quiet to moderately active geomagnetic activity.

Overall, the paper is well-written, the reasoning is clearly exposed, and it reaches substantial conclusions. Below are a few specific comments which could be considered to improve the manuscript before final publication.

**Specific comments (major)**

(I) It would be good to have a figure in the same format as Figure 3 showing the number of events in each bin, both for azimuthal and radial oscillating events. Indeed, the text mentions on several occasions that a given bin should be considered with caution due to the small number of events it contains. In particular, such an additional figure could help in analysing Figure 4, as one might want to be cautious in drawing conclusions with so many blank bins in this figure.

(II) l. 327: "solar wind dynamic pressure (Pd) and the IMF Bz values were also examined (not shown)." → If the results on the dynamic pressure are mentioned in the conclusion (see l. 386), I would recommend to show the analysis of these parameters in the paper.

**Specific comments (minor)**

- l. 42: I suggest to give the frequency range for ULF waves from the very beginning of the introduction.

- l. 69: The full name of THEMIS should be given when mentioned for the first time after the abstract (see AnGeo guidelines: https://www.annales-geophysicae.net/for_authors/manuscript_preparation.html).

- l. 74: Please define MLT.

- l. 76 and 78: "Susumu Kokubun (2013)" → please check the reference. I think it should be "Kokubun (2013)", and in the reference list it should appear as "Kokubun, S. (2013). ULF waves in the outer magnetosphere...".

- l. 82: Pi2 should be defined.

- l. 108: It could be worth briefly defining the GSM coordinate system here.

- l. 120: I assume Dp stands for "dynamic pressure"; perhaps it would be better to make it fully clear to the reader.

- Figure 2: I would suggest that, for a given event (A/B), the three components of the velocity, magnetic field, electric field should have the same y-axis limits. This would better emphasise the azimuthal or radial nature of the oscillations. This could also be done for the components of the PSD of the velocity, as the relative importance of the peaks would be immediately visible, without needing to look at the scale for comparison. An exception should obviously be made for panel (d2), as the Bz range for event B is very large.

- l. 174: "calculated by dividing the total event times by the total observation times" → do you mean the total *number* of events/observations? Or the total *duration* of events/observations? This wording may be ambiguous; please rephrase.

- l. 177: "For azimuthal oscillating events, there is no clear dawn-dusk asymmetry in the occurrence rates" → based on Figure 3, this statement seems a bit too strong, since high occurrence rates (red colour) span within 18–20 MLT on the duskside vs 5–6 MLT on the dawnside. Perhaps rephrasing this statement into something like "For azimuthal oscillating events, the dawn-dusk asymmetry in the occurrence rates is less clear than for radial oscillating events" would be better. Similarly, this statement should also be made less strong in the conclusion and abstract.

- l. 202: "the frequency can be as low as 0.55 mHz" which is the lower frequency in the Pc6 band retained in this study. Could there be waves with even lower frequency observed? To my knowledge, there is no upper limit in the period of Pc6 oscillation (see Saito, T. (1978), Long-period irregular magnetic pulsation, Pi3, Space Sci. Rev., 21(4), 427–467, doi:10.1007/BF00173068), so it could be interesting to check whether pulsations of even lower frequency can be identified.

- l. 226: "The second row shows the 1.26-3.26 mHz (Fig. 6a) and 2.03-4.03 mHz (Fig. 6b) band-pass filtered Ba and Er components" → how were those frequency bands selected?

- l. 319-320: The calculation of the normalised event number does not seem fully clear to me. Is it so that the number of events is divided by the proportion of solar wind speed within a given bin to the total duration of solar wind measurements? Or are events binned according to the mean/median solar wind speed at the time when they were observed? Please rephrase this explanation.

**Copyediting and typesetting**

- l. 57: "primarily" being an adverb, it cannot be used in this context. Instead, one could write, for instance, "Pc5 and Pc6 waves are the most common waves observed at high latitudes and in the magnetotail."

- l. 60: "nightside" is generally written in a single word (same for dawnside, dayside, duskside...)

- l. 67: "the both occurrence and frequency distributions..." –> "both the occurrence and frequency distributions..."

- l. 102, l. 201: earth → Earth

- l. 106: sub-solar → subsolar

- l. 108: "whose X axis **is** rotated"

- l. 145: "to satisfy **the** criteria mentioned above"

- l. 154: "of **the** three components"

- l. 260: "the magnetic field line**s** in the nightside **are** very stretched"

- l. 276: "K-H instabilities are **more** inclined to occur in the dawnside than in the duskside"

- l. 278: "more events are needed **to** further study the definite reasons"

- l. 286: Alfven → Alfvén

- l. 321: "the ULF waves occurrences increase with" → "the ULF wave occurrence increases with"

- l. 322: a sources → a source

- l. 324: "the waves occurrences are higher" → "the wave occurrence is higher"

- l. 327: occurrence

- l. 336: (Forsyth et al., 2015) → Forsyth et al. (2015)

- l. 379: "the peak frequency decreases **with increasing** radial distance"

- l. 383: "the frequencies for all the events in this paper do not show obvious dawn-dusk asymmetry **contrary to results from** previous studies for waves in the inner magnetosphere" (the original phrasing can be ambiguous and interpreted the other way round)

- l. 387: maybe → may be
- l. 389: "that the ULF waves **are** most likely to occur"

---

## Referee Comment (RC2) · Anonymous Referee #2 · 11 Jun 2018

This paper presents a statistical analysis of ULF waves in the nightside magnetosphere including the tail region up to geocentric distances of 32 Re. The wave events are identified using plasma bulk velocity data to capture fundamental field line oscillations. The wave events are classified into radial and azimuthal type and standing and nonstanding modes. The most important result is the finding that standing waves are hardly detected beyond 16 Re. I find this result to be very interesting and significant. I recommend publication of the manuscript after the authors have considered minor comments listed below.

Line 78. Please change "Susumu Kokubun" to "Kokubun"

Line 136. "quasi monochromatic" Is this judgement made by visual inspection of the time series data? If so, it may explain some discrepancies between the present and

previous studies (see my comment on line 267 below).

Line 169, Figure 2. I recommend using a common amplitude scale for the components of each vector quantity. It makes easier to visually grasp the amplitude difference between the toroidal and poloidal components.

Line 171. The Ez component is not zero. This should be pointed out in the main text and explained in relation to how you define the field line coordinate system.

Line 232. The phase difference can be $\sim$-90 degrees, depending on the magnetic latitude (or distance from the tail midplane where Bx = 0). I hope that events exhibiting this phase delay are also included.

Line 239. I guess that this probability means the probability that a given azimuthal or radial wave event shows signatures of a standing wave, not the probability that you find a standing wave at a given time. Please clarify.

Line 240. Change "deep" to something like "dark"

Line 259. "fundamental eigenmode". You can justify this mode identification by examining the relationship between the sign of the Er-Ba phase difference and the magnetic latitude of the spacecraft. You can also identify second harmonic from the phase difference. Have you found any second harmonic waves? Second harmonic poloidal (radial) waves have been reported at R < 10 (e.g., Hughes et al., 1979), and I wonder if you have encountered any at R > 10.

Hughes, W. J., McPherron, R. I., Barfield, J. N., & Mauk, B. H. (1979). A compressional Pc4 pulsation observed by three satellites in geostationary orbit near local midnight. Planetary and Space Science, 27(6), 821-840. doi:10.1016/0032-0633(79)90010-2

Line 267. This is different from the Geotail result obtained by Kokubun (2013, Figure 15) and Takahashi et al. (2014, doi:10.1002/2014ja020274, Figure 5). Please comment on this difference and offer explanation if possible.

[Figure]

Line 287. "Poulters" means "Allan and Poulter?"

Line 298, "Highly stretched field lines". Is it possible that some events are observed on open field lines?

Line 331. Figure 8. It would be better if the vertical axis shows the probability of detecting a wave event (instead of the number of events) in each bin for the solar wind velocity and AE index.

---

## Author Comment (AC2) · 25 Jul 2018

First of all, we would like to thank the reviewer for the helpful comments and suggestions. We are very appreciative of that. Our responses to the comments are in blue font. In the revised manuscript, the modifications are highlighted in red font.

This paper presents a statistical analysis of ULF waves in the nightside magnetosphere including the tail region up to geocentric distances of 32 Re. The wave events are identified using plasma bulk velocity data to capture fundamental field line oscillations. The wave events are classified into radial and azimuthal type and standing and nonstanding modes. The most important result is the finding that standing waves are

hardly detected beyond 16 Re. I find this result to be very interesting and significant. I recommend publication of the manuscript after the authors have considered minor comments listed below.

Line 78. Please change "Susumu Kokubun" to "Kokubun"

Revised.

Line 136. "quasi monochromatic" Is this judgement made by visual inspection of the time series data? If so, it may explain some discrepancies between the present and previous studies (see my comment on line 267 below).

Thanks for your suggestions. In this study, the "Quasi monochromatic" is not exactly made by visual inspection of the time series data. We conduct Fast Fourier Translation (FFT) analysis to all the candidate time series as to the event shown in Fig. 2(l-n). Then, we judge whether there is an obvious spectral peak in FFT spectrum by visual inspection. Only events with an obvious single spectral peak are considered as "quasi monochromatic" wave. We added one sentence in lines 158-159.

Line 169, Figure 2. I recommend using a common amplitude scale for the components of each vector quantity. It makes easier to visually grasp the amplitude difference between the toroidal and poloidal components.

Ok. Figure 2 has been re-plotted by using the same y-axis limit for each component in the revised manuscript.

Line 171. The Ez component is not zero. This should be pointed out in the main text and explained in relation to how you define the field line coordinate system.

Thanks for your suggestion. The sentence has been added in lines 163-166 in the revised manuscript: "Note that the magnetic field vector used for calculating **E (E =** $\delta$**V** $\times$ **B)** at each moment may be deviate from the z-axis determined by 30 minutes sliding average of the magnetic field data. Therefore, the Ez component will have a

little deviation from zero in the FAC coordinate system as shown in Fig. 2g.".

Line 232. The phase difference can be ~-90 degrees, depending on the magnetic latitude (or distance from the tail midplane where Bx = 0). I hope that events exhibiting this phase delay are also included.

Thanks for your comments. The events with $-90°$ phase difference have already been included in our list of events. Because either the phase deference of $\sim90°$ or $-90°$ is considered as the characteristic of a standing wave. We rephrased the words "the phase differences" as "the absolute value of the phase differences" in lines 238 and 243 in the revised manuscript.

Line 239. I guess that this probability means the probability that a given azimuthal or radial wave event shows signatures of a standing wave, not the probability that you find a standing wave at a given time. Please clarify.

Yes, the sentence has been rephrased as "Figure 7 shows the radial distribution of the probability that a given azimuthal or radial oscillating wave event shows signatures of a standing wave." in lines 249-250 in the revised manuscript.

Line 240. Change "deep" to something like "dark"

Revised. See line 250 in the revised manuscript.

Line 259. "fundamental eigenmode". You can justify this mode identification by examining the relationship between the sign of the Er-Ba phase difference and the magnetic latitude of the spacecraft. You can also identify second harmonic from the phase difference. Have you found any second harmonic waves? Second harmonic poloidal (radial) waves have been reported at R < 10 (e.g., Hughes et al., 1979), and I wonder if you have encountered any at R > 10.

Hughes, W. J., McPherron, R. I., Barfield, J. N., & Mauk, B. H. (1979). A compressional Pc4 pulsation observed by three satellites in geostationary orbit near local midnight. Planetary and Space Science, 27(6), 821-840. doi:10.1016/0032-0633(79)90010-2

Thanks for your suggestions. According to your suggestion, we try to identify the second harmonic waves by the E-B phase difference and the magnetic latitude. Figure S1 is a schematic diagram describing the latitude dependence of phase difference for the fundamental waves and second harmonic waves. The green dashed line P1 (P2) indicates a fixed point slightly on the north (south) side of the magnetic equator plane. Panel b (c) shows the temporal variation of B, V and E at the P1 (P2) point. We can see that the sign of E-B phase difference between the fundamental waves and second harmonic waves is opposite at the same observation latitudes. The azimuthal oscillating wave could be second harmonic wave if Er lead (lag) Ba by $\sim 90°$ when MLAT<0 (MLAT>0), and the radial oscillating wave could be second harmonic if Ea lead (lag) Br by $\sim 90°$ when MLAT>0 (MLAT<0). In total, we find that about 3.03% (33) azimuthal oscillating wave events may be second harmonic waves and 2.65% (15) radial oscillating wave events may be second harmonic waves, as shown in Figure S2. The above analysis result is not contradictory to our opinion that most of our standing wave events are fundamental waves. We added two sentences about the harmonic mode in lines 270-272 in the revised manuscript.

Line 267. This is different from the Geotail result obtained by Kokubun (2013, Figure 15) and Takahashi et al. (2014, doi:10.1002/2014ja020274, Figure 5). Please comment on this difference and offer explanation if possible.

Kokubun (2013, Figure 15) shows that the number of waves is higher on the dawnside than duskside, at first glance, it is different from our results. We noticed that, in his work, the orbit normalization was not conducted and the events they focused on mainly occur in the dayside. The number of events (153) observed in the nightside is less in his work, which is probably because the amplitude of waves is required to be greater than 30 km/s in his work, while the amplitude of waves should be greater than 25 km/s in our study.

Takahashi et al. (2014, Figure 5) shows that the occurrence rate of toroidal waves is higher on the dawnside than duskside and the occurrence rate is zero in the midnight

region, which are different from our results. The possible reason is that they only focused on the pure toroidal wave, while azimuthal oscillating waves with comparable power in Va and Vr are also included in our list of events. Thus, more azimuthal oscillating waves could be observed in the duskside in our study, because of the possible coupling between azimuthal oscillating waves and radial oscillating waves (with higher occurrence in the dusk sector). Another possible reason for these differences is that their waves were identified automatically by program using a 60 min data window and there was no precise beginning and ending time. A quantitative standard is used in our study to determine the beginning and ending time of each event, which may lead to a different list of events between our than their works. We added some additional comments in lines 287-292 in the revised manuscript.

Line 287. "Poulters" means "Allan and Poulter?"

Yes, revised.

Line 298, "Highly stretched field lines". Is it possible that some events are observed on open field lines?

It is an interesting question. However, we can only say statistically that the events we studied were happened on closed magnetic field lines. We have calculated the elevation angle of the magnetic field line for all wave events as shown in Fig. S3. It found that 61.70% of the events have an elevation angle larger than $45°$ and only 2.48% events with an elevation angle $<10°$. It suggests that most of our events are observed near the magnetic equatorial plane. In addition, we mainly use ion velocity data to identify ULF waves, which are usually reliably measured in the plasma sheet. Therefore, it is unlikely that the events studied in this work were occurred on open magnetic field lines. We added some additional comments in lines 269 and 273-277 in the revised manuscript.

Line 331. Figure 8. It would be better if the vertical axis shows the probability of detecting a wave event (instead of the number of events) in each bin for the solar wind

velocity and AE index.

Thanks for your suggestion. we have re-plotted Fig. 8. The new vertical axis shows the probability of detecting a wave event instead of the normalized number of events in each bin. The detailed calculation formula for the probability is as follows (take the solar wind Vx as an example):

$$Normalized\ event\ number = \frac{the\ number\ of\ events\ in\ a\ given\ bin}{the\ duration\ proportion\ of\ solar\ wind\ Vx\ in\ a\ given\ bin}$$

$$= \frac{the\ number\ of\ events\ in\ a\ given\ bin}{\frac{duration\ of\ background\ Vx\ in\ a\ given\ bin}{total\ duration\ of\ background\ Vx}}$$

$$Probability = \frac{the\ normalized\ event\ number\ in\ a\ given\ bin}{the\ total\ normalized\ event\ number\ of\ all\ bins}$$

The sentence has been rephrased to "The Y-axis indicates the probability of detecting one wave event in each bin. The background solar wind data is obtained from OMNI from 2008 to 2015." in lines 338-339 in the revised manuscript.

Please also note the supplement to this comment:
https://www.ann-geophys-discuss.net/angeo-2018-39/angeo-2018-39-AC2-supplement.pdf

[Figure]

[Figure]

**Fig. 1.** Figure S1. Schematic illustration of magnetic field, velocity and electric field for the fundamental wave and second harmonic waves in the magnetic meridian plane. Refer to Fig. 3 of Takahashi et al.

[Figure]

**Fig. 2.** Figure S2. The distribution of second harmonic waves in the GSM* X-Y plane.

[Figure]

**Fig. 3.** Figure S3. The distribution of elevation angle of the magnetic field line.

**Supplement:**

[revised manuscript text omitted]

---

## Author Response (AR1)

**● A point-by-point response to the reviews**

**First of all, we would like to thank the reviewers for the helpful comments and suggestions. We are very appreciative of that. Our responses to the comments are in bold font. In the revised manuscript, the modifications are highlighted in red font.**

Anonymous Referee #1

General comments

This paper presents results from a statistical study of ULF waves in the Pc5-Pc6 frequency range in the magnetotail using magnetic field and plasma data from instruments onboard the THEMIS satellite during 2008–2015. Azimuthally oscillating waves and radially oscillating waves are studied separately. The authors found that in the near-Earth magnetotail the dawn-dusk asymmetry of wave occurrence is observed only in the case of radially oscillating waves. In the far magnetotail, they reveal a higher occurrence rate during post-midnight magnetic local times than during pre-midnight, and that the peak frequency of waves decreases with increasing radial distance from Earth. A majority of events in the near-Earth magnetotail are standing waves, while these represent very small percentage of the events in the far magnetotail. No dawn-dusk asymmetry could be found in the wave frequencies. Finally, the authors studied the effects of solar wind parameters and geomagnetic activity, and found that ULF wave occurrence is favoured by high solar wind speed, high solar wind pressure variability, but quiet to moderately active geomagnetic activity.

Overall, the paper is well-written, the reasoning is clearly exposed, and it reaches substantial conclusions. Below are a few specific comments which could be considered to improve the manuscript before final publication.

Specific comments (major)

(I) It would be good to have a figure in the same format as Figure 3 showing the number of events in each bin, both for azimuthal and radial oscillating events. Indeed, the text mentions on several occasions that a given bin should be considered with caution due to the small number of events it contains. In particular, such an additional figure could help in analysing Figure 4, as one might want to be cautious in drawing conclusions with so many blank bins in this figure.

**We have added the distribution of event numbers in Fig. 3 in the revised manuscript. Two sentences**

have been added in lines 173-175: "Figure 3a and 3b shows the spatial distribution of the number of events in the GSM* X-Y plane, both for azimuthal (left panel) and radial (right panel) oscillating wave events. The blank bins inside the magnetopause indicate that there are no events.".

(II) l. 327: "solar wind dynamic pressure (Pd) and the IMF Bz values were also examined (not shown)." ! If the results on the dynamic pressure are mentioned in the conclusion (see l. 386), I would recommend to show the analysis of these parameters in the paper.

**Thanks for your suggestions. The relationship between the occurrence of ULF waves and the relative variation of solar wind dynamic pressure (Pd) and the IMF Bz values have been added in Fig. 8 (Fig. 8c and 8d) in the revised manuscript. And the calculation formula of the relative variation of Pd have been added in lines 347-349 in the revised manuscript.**

Specific comments (minor)

• l. 42: I suggest to give the frequency range for ULF waves from the very beginning of the introduction.

**The frequency range for ULF waves have been added in line 42 in the revised manuscript.**

• l. 69: The full name of THEMIS should be given when mentioned for the first time after the abstract (see AnGeo guidelines: https://www.annales-geophysicae.net/for_authors/manuscript_preparation.html).

**Ok. The full name of THEMIS has been added in lines 69-70 in the revised manuscript.**

• l. 75: Please define MLT.

**The definition of MLT have been added in line 75 in the revised manuscript.**

• l. 77 and 79: "Susumu Kokubun (2013)" ! please check the reference. I think it should be "Kokubun (2013)", and in the reference list it should appear as "Kokubun, S. (2013). ULF waves in the outer magnetosphere...".

**Yes, revised.**

• l. 82: Pi2 should be defined.

**Ok. The definition of Pi2 have been added in line 83 in the revised manuscript.**

• l. 108: It could be worth briefly defining the GSM coordinate system here.

**Ok. The definition of GSM coordinate system has been added in lines 115-119 in the revised manuscript.**

• l. 120: I assume Dp stands for "dynamic pressure"; perhaps it would be better to make it fully clear to the reader.

**Yes, it stands for "dynamic pressure". The full name of Dp has been added in line 112 in the revised manuscript.**

• Figure 2: I would suggest that, for a given event (A/B), the three components of the velocity, magnetic field, electric field should have the same y-axis limits. This would better emphasise the azimuthal or radial nature of the oscillations. This could also be done for the components of the PSD of the velocity, as the relative importance of the peaks would be immediately visible, without needing to look at the scale for comparison. An exception should obviously be made for panel (d2), as the Bz range for event B is very large.

**Ok. We re-plotted Fig. 2 using the same y-axis limits for the components of each vector quantity (except Bz) in the revised manuscript.**

• l. 174: "calculated by dividing the total event times by the total observation times" ! do you mean the total number of events/observations? Or the total duration of events/observations? This wording may be ambiguous; please rephrase.

**Ok. The sentence has been rephrased as "calculated by dividing the total duration of all events by the total duration of observations in each bin" in line 183 in the revised manuscript.**

• l. 177: "For azimuthal oscillating events, there is no clear dawn-dusk asymmetry in the occurrence rates" ! based on Figure 3, this statement seems a bit too strong, since high occurrence rates (red colour) span within 18–20 MLT on the duskside vs 5–6 MLT on the dawnside. Perhaps rephrasing this statement into something like "For azimuthal oscillating events, the dawn-dusk asymmetry in the occurrence rates is less clear than for radial oscillating events" would be better. Similarly, this statement should also be made less strong in the conclusion and abstract.

**Thank you, the sentence has been rephrased as "For radial oscillating events, the occurrence rates of waves are higher on the duskside than dawnside. For azimuthal oscillating events, the dawn-dusk asymmetry in the occurrence rates is less clear than that for radial oscillating events." in lines 187-189 in the revised manuscript. Similarly, the sentence has also been rephrased in lines 281-283 and 397-399 in the revised manuscript.**

• l. 202: "the frequency can be as low as 0.55 mHz" which is the lower frequency in the Pc6 band retained in this study. Could there be waves with even lower frequency observed? To my knowledge, there is no upper limit in the period of Pc6 oscillation (see Saito, T. (1978), Long-period irregular magnetic pulsation, Pi3, Space Sci. Rev., 21(4), 427–467, doi:10.1007/BF00173068), so it could be interesting to check whether pulsations of even lower frequency can be identified.

**The field aligned coordinates used in this work are obtained by subtracting the 30min sliding average background magnetic field. So, we only focus on waves with frequencies above 0.55mHz. However, in the preliminary screening stage, we did notice two wave events with frequencies as low as 0.49 mHz and 0.51mHz, which can be studied in the future work. Besides, the words "Pc6 (>600s)" were revised in line 57 in the revised manuscript. The paper of Saito (1978) was quoted in the lines 58 and 496-497 in the revised manuscript.**

- l. 226: "The second row shows the 1.26-3.26 mHz (Fig. 6a) and 2.03-4.03 mHz (Fig. 6b) band-pass filtered Ba and Er components"!how were those frequency bands selected?

**The lower (upper) limit of the frequency bands is obtained by subtracting (adding) 1 mHz from the peak frequency in Fig. 2(l-n). We added one sentence in lines 237-238.**

- l. 319-320: The calculation of the normalised event number does not seem fully clear to me. Is it so that the number of events is divided by the proportion of solar wind speed within a given bin to the total duration of solar wind measurements? Or are events binned according to the mean/median solar wind speed at the time when they were observed? Please rephrase this explanation.

**Yes, the detailed calculation formula for the normalized event number is as follows (take that dividing by the duration proportion of solar wind Vx as an example):**

$$\text{Normalized event number} = \frac{\text{the number of events in a given bin}}{\text{the duration proportion of solar wind Vx in a given bin}}$$

$$= \frac{\text{the number of events in a given bin}}{\dfrac{\text{duration of background Vx in a given bin}}{\text{total duration of background Vx}}}$$

**Taking into account the suggestion of another reviewer, we re-plotted Figure 8. The new vertical axis shows the probability of detecting a wave event instead of the normalized number of events in each bin. The probability is calculated by dividing the normalized event number in a given bin by the total normalized event number of all bins (take the solar wind Vx as an example). The sentence has been rephrased to "The Y-axis indicates the probability of detecting one wave event in each bin. The background solar wind data is obtained from OMNI from 2008 to 2015." in lines 338-339 in the revised manuscript.**

Copyediting and typesetting

**All the following problems have been corrected in the revised manuscript accordingly.**

- l. 57: "primarily" being an adverb, it cannot be used in this context. Instead, one could write, for instance, "Pc5 and Pc6 waves are the most common waves observed at high latitudes and in the

magnetotail."

- l. 60: "nightside" is generally written in a single word (same for dawnside, dayside, duskside...)

- l. 67: "the both occurrence and frequency distributions..." –> "both the occurrence and frequency distributions..."

- l. 102, l. 201: earth ! Earth

- l. 106: sub-solar ! subsolar

- l. 108: "whose X axis **is** rotated"

- l. 145: "to satisfy **the** criteria mentioned above"

- l. 154: "of **the** three components"

- l. 260: "the magnetic field line**s** in the nightside **are** very stretched"

- l. 276: "K-H instabilities are **more** inclined to occur in the dawnside than in the duskside"

- l. 278: "more events are needed **to** further study the definite reasons"

- l. 286: Alfven ! Alfvén

- l. 321: "the ULF waves occurrences increase with" ! "the ULF wave occurrence increases with"

- l. 322: a sources ! a source

- l. 324: "the waves occurrences are higher" ! "the wave occurrence is higher"

- l. 327: occurrence

- l. 336: (Forsyth et al., 2015) ! Forsyth et al. (2015)

- l. 379: "the peak frequency decreases **with increasing** radial distance"

- l. 383: "the frequencies for all the events in this paper do not show obvious dawndusk asymmetry **contrary to results from** previous studies for waves in the inner magnetosphere" (the original phrasing can be ambiguous and interpreted the other way round)

- l. 387: maybe ! may be

- l. 389: "that the ULF waves **are** most likely to occur"

Anonymous Referee #2

This paper presents a statistical analysis of ULF waves in the nightside magnetosphere including the tail region up to geocentric distances of 32 Re. The wave events are identified using plasma bulk velocity data to capture fundamental field line oscillations. The wave events are classified into radial and azimuthal type and standing and nonstanding modes. The most important result is the finding that standing waves are hardly detected beyond 16 Re. I find this result to be very interesting and significant. I recommend publication of the manuscript after the authors have considered minor comments listed below.

Line 78. Please change "Susumu Kokubun" to "Kokubun"

**Revised.**

Line 136. "quasi monochromatic" Is this judgement made by visual inspection of the time series data? If so, it may explain some discrepancies between the present and previous studies (see my comment on line 267 below).

**Thanks for your suggestions. In this study, the "Quasi monochromatic" is not exactly made by visual inspection of the time series data. We conduct Fast Fourier Translation (FFT) analysis to all the candidate time series as to the event shown in Fig. 2(l-n). Then, we judge whether there is an obvious spectral peak in FFT spectrum by visual inspection. Only events with an obvious single spectral peak are considered as "quasi monochromatic" wave. We added one sentence in lines 158-159.**

Line 169, Figure 2. I recommend using a common amplitude scale for the components of each vector quantity. It makes easier to visually grasp the amplitude difference between the toroidal and poloidal components.

**Ok. Figure 2 has been re-plotted by using the same y-axis limit for each component in the revised manuscript.**

Line 171. The Ez component is not zero. This should be pointed out in the main text and explained in relation to how you define the field line coordinate system.
**Thanks for your suggestion. The sentence has been added in lines 163-166 in the revised manuscript: "Note that the magnetic field vector used for calculating E (E = −δV×B) at each moment may be**

deviate from the z-axis determined by 30 minutes sliding average of the magnetic field data. Therefore, the Ez component will have a little deviation from zero in the FAC coordinate system as shown in Fig. 2g.".

Line 232. The phase difference can be ~-90 degrees, depending on the magnetic latitude (or distance from the tail midplane where Bx = 0). I hope that events exhibiting this phase delay are also included.

**Thanks for your comments. The events with -90° phase difference have already been included in our list of events. Because either the phase deference of ~90° or -90° is considered as the characteristic of a standing wave. We rephrased the words "the phase differences" as "the absolute value of the phase differences" in lines 238 and 243 in the revised manuscript.**

Line 239. I guess that this probability means the probability that a given azimuthal or radial wave event shows signatures of a standing wave, not the probability that you find a standing wave at a given time. Please clarify.

**Yes, the sentence has been rephrased as "Figure 7 shows the radial distribution of the probability that a given azimuthal or radial oscillating wave event shows signatures of a standing wave." in lines 249-250 in the revised manuscript.**

Line 240. Change "deep" to something like "dark"

**Revised. See line 250 in the revised manuscript.**

Line 259. "fundamental eigenmode". You can justify this mode identification by examining the relationship between the sign of the Er-Ba phase difference and the magnetic latitude of the spacecraft. You can also identify second harmonic from the phase difference. Have you found any second harmonic waves? Second harmonic poloidal (radial) waves have been reported at R < 10 (e.g., Hughes et al., 1979), and I wonder if you have encountered any at R > 10.

Hughes, W. J., McPherron, R. I., Barfield, J. N., & Mauk, B. H. (1979). A compressional Pc4 pulsation observed by three satellites in geostationary orbit near local midnight. Planetary and Space Science, 27(6), 821-840. doi:10.1016/0032-0633(79)90010-2

**Thanks for your suggestions. According to your suggestion, we try to identify the second harmonic waves by the E-B phase difference and the magnetic latitude. Figure S1 is a schematic diagram describing the latitude dependence of phase difference for the fundamental waves and second harmonic waves. We can see that the sign of E-B phase difference between the fundamental waves and second harmonic waves is opposite at the same observation latitudes. The azimuthal oscillating wave could be second harmonic wave if Er lead (lag) Ba by ~90° when MLAT<0 (MLAT>0), and the radial oscillating wave could be second harmonic if Ea lead (lag) Br by ~90° when MLAT>0 (MLAT<0). In total, we find that about 3.03% (33) azimuthal oscillating wave events may be second**

harmonic waves and 2.65% (15) radial oscillating wave events may be second harmonic waves, as shown in Figure S2. The above analysis result is not contradictory to our opinion that most of our standing wave events are fundamental waves. We added two sentences about the harmonic mode in lines 270-272 in the revised manuscript.

[Figure]

Figure S1. (a1) Schematic illustration of magnetic field B (blue line) and velocity V (red arrow) for the fundamental azimuthal oscillating wave in the magnetic meridian plane. The black dashed line indicates the magnetic equator plane, and the green dashed line P1 (P2) indicates a fixed point slightly on the north (south) side of the magnetic equator plane. (b1) The temporal variation of Ba, Va and Er at the P1 point. (c1) The temporal variation of Br, Vr and Ea at the P2 point. (a2-c2) are the same as (a1-c1), but for the fundamental radial oscillating wave. (a3-c3) and (a4-c4) are the

**same as (a1-c1) and (a2-c2), but for the second harmonic oscillating waves. Refer to Fig. 3 of Takahashi et al. (2014).**

[Figure]

**Figure S2. The distribution of second harmonic waves in the GSM* X-Y plane.**

Line 267. This is different from the Geotail result obtained by Kokubun (2013, Figure 15) and Takahashi et al. (2014, doi:10.1002/2014ja020274, Figure 5). Please comment on this difference and offer explanation if possible.

**Kokubun (2013, Figure 15) shows that the number of waves is higher on the dawnside than duskside, at first glance, it is different from our results. We noticed that, in his work, the orbit normalization was not conducted and the events they focused on mainly occur in the dayside. The number of events (153) observed in the nightside is less in his work, which is probably because the amplitude of waves is required to be greater than 30 km/s in his work, while the amplitude of waves should be greater than 25 km/s in our study.**

**Takahashi et al. (2014, Figure 5) shows that the occurrence rate of toroidal waves is higher on the dawnside than duskside and the occurrence rate is zero in the midnight region, which are different from our results. The possible reason is that they only focused on the pure toroidal wave, while azimuthal oscillating waves with comparable power in Va and Vr are also included in our list of events. Thus, more azimuthal oscillating waves could be observed in the duskside in our study, because of the possible coupling between azimuthal oscillating waves and radial oscillating waves**

(with higher occurrence in the dusk sector). **Another possible reason for these differences is that their waves were identified automatically by program using a 60 min data window and there was no precise beginning and ending time. A quantitative standard is used in our study to determine the beginning and ending time of each event, which may lead to a different list of events between our than their works. We added some additional comments in lines 287-292 in the revised manuscript.**

Line 287. "Poulters" means "Allan and Poulter?"

**Yes, revised.**

Line 298, "Highly stretched field lines". Is it possible that some events are observed on open field lines?

**It is an interesting question. However, we can only say statistically that the events we studied were happened on closed magnetic field lines. We have calculated the elevation angle of the magnetic field line for all wave events as shown in Fig. S3. It found that 61.70% of the events have an elevation angle larger than 45° and only 2.48% events with an elevation angle <10°. It suggests that most of our events are observed near the magnetic equatorial plane. In addition, we mainly use ion velocity data to identify ULF waves, which are usually reliably measured in the plasma sheet. Therefore, it is unlikely that the events studied in this work were occurred on open magnetic field lines. We added some additional comments in lines 269 and 273-277 in the revised manuscript.**

[Figure]

**Figure S3. The distribution of elevation angle of the magnetic field line.**

Line 331. Figure 8. It would be better if the vertical axis shows the probability of detecting a wave event (instead of the number of events) in each bin for the solar wind velocity and AE index.

Thanks for your suggestion. we have re-plotted Fig. 8. The new vertical axis shows the probability of detecting a wave event instead of the normalized number of events in each bin. The detailed calculation formula for the probability is as follows (take the solar wind Vx as an example):

$$\text{Normalized event number} = \frac{\text{the number of events in a given bin}}{\text{the duration proportion of solar wind Vx in a given bin}}$$

$$= \frac{\text{the number of events in a given bin}}{\frac{\text{duration of background Vx in a given bin}}{\text{total duration of background Vx}}}$$

$$\text{Probability} = \frac{\text{the normalized event number in a given bin}}{\text{the total normalized event number of all bins}}$$

The sentence has been rephrased to "The Y-axis indicates the probability of detecting one wave event in each bin. The background solar wind data is obtained from OMNI from 2008 to 2015." in lines 338-339 in the revised manuscript.

**● A list of all relevant changes made in the manuscript**

**Line 42. The frequency range for ULF waves was added.**

**Line 57. "Pc6 (600-1800s)" ! "Pc6 (>600s)". "primarily" ! "most common"**

**Line 58. The paper of Saito (1978) was quoted.**

**Line 60. "dusk side" ! "duskside", "dawn side" ! "dawnside", "night side" ! "nightside" in the full text.**

**Line 67. "the both" ! "both the".**

**Line 69. The full name of THEMIS was added.**

**Line 75. The definition of MLT was added.**

**Line 77. "Susumu Kokubun, 2013" ! "Kokubun, 2013".**

**Line 79. "Susumu Kokubun (2013)" ! "Kokubun (2013)"**

**Line 82. The definition of Pi2 was added.**

**Line 103. "earth" ! "Earth".**

**Line 107. "sub-solar" ! "subsolar".**

**Line 112. The full name of Dp was added. "npa"! "nPa".**

**Line 115.** "whose X axis **is** rotated".

**Lines 115-119. The definition of GSM coordinate system was added**

**Line 148.** "to satisfy **the** criteria mentioned above".

**Line 157.** "of **the** three components".

**Line 158. The explanation of "quasi monochromatic" was added.**

**Line 163. The explanation of "The Ez component is not zero" was added.**

**Line 167. We re-plotted Fig. 2 using the same y-axis limits.**

**Line 173. We added the description of Fig. 3a and 3b.**

**Line 183. "event times" ! "duration of all events", "observation times" ! "duration of observations in each bin"**

**Line 184. We added the description of blank bins in Fig. 3c and 3d.**

**Line 187. We rephrased the statement about the distribution of occurrence rates of azimuthal oscillating waves in the near-earth region. This statement was also rephrased in lines 281 and 397.**

**Line 194. We re-plotted Fig. 3 and added the distribution of event numbers.**

**Line 211. "earth" ! "Earth"**

**Line 237. The method of selecting the frequency bands was added.**

**Lines 238 &243. "the phase differences" ! "the absolute value of the phase differences"**

**Line 249. We rephrased the description of the Y-axis of Fig. 7.**

**Line 250. "deep" ! "dark".**

**Lines 269-277. A few lines about "the harmonic mode" and "open magnetic field lines" were added.**

**Line 287. We added the differences between our result and that of Takahashi et al. (2014), and discussed the potential reasons.**

**Line 275.** "the magnetic field line**s** in the nightside **are** very stretched".

**Line 296.** "are **more** inclined to occur".

**Line 299.** "needed **to** further study".

**Line 306. "Alfven" ! "Alfvén", "Poulters" ! "Allan and Poulter"**

**Line 338. We rephrased the description of Y-axis of Figure 8.**

**Line 340. "the ULF waves occurrences increase with" ! "the ULF wave occurrence increases with"**

**Line 341. "a sources" ! "a source".**

**Line 342. "the waves occurrences are higher" ! "the wave occurrence is higher".**

**Line 346. "occurrences" ! "occurrence".**

**Line 347. The calculation formula of the relative variation of Pd was added.**

**Line 351. We re-plotted Fig. 8. The relationship between the occurrence of ULF waves and the relative variation of Pd and the IMF Bz values were added. The new Y-axis shows the probability of detecting a wave event instead of the normalized number of events.**

**Line 357. "(Forsyth et al., 2015)" ! "Forsyth et al. (2015)".**

**Line 401. "decreases as the increase of radial distance" ! "decreases with increasing radial distance".**

**Line 406. "as found in previous studies" ! "contrary to results from previous studies".**

**Line 410. "maybe" ! "may be".**

**Line 412.** "the ULF waves **are** most likely to occur"

**Line 496. The paper of Saito (1978) was quoted.**

**Line 526. " Susumu, K" ! "Kokubun, S"**

**Line 532. The paper of Takahashi (2014) was quoted.**

● **A marked-up manuscript version**

[revised manuscript text omitted]